# Kinetic Aspects of Ethylene Glycol Degradation Using UV-C Activated Hydrogen Peroxide (H_2_O_2_/UV-C)

**DOI:** 10.3390/molecules30010049

**Published:** 2024-12-26

**Authors:** Timur Fazliev, Mikhail Lyulyukin, Denis Kozlov, Dmitry Selishchev

**Affiliations:** 1Research and Educational Center “Institute of Chemical Technologies”, Novosibirsk State University, Pirogova St. 2, Novosibirsk 630090, Russia; t.fazliev@g.nsu.ru (T.F.); m.lyulyukin@g.nsu.ru (M.L.); d.kozlov@g.nsu.ru (D.K.); 2Competence Center of the National Technological Initiative “Hydrogen as the Basis of a Low-Carbon Economy”, Lavrentieva Ave. 7, Novosibirsk 630090, Russia

**Keywords:** environmental protection, wastewater treatment, ethylene glycol (EG), advanced oxidation processes (AOPs), photodegradation, H_2_O_2_/UV-C

## Abstract

Ethylene glycol (EG) is a contaminant in the wastewater of airports because it is commonly used in aircraft deicing fluids during the cold season in northern regions. Ethylene glycol by itself has relatively low toxicity to mammals and aquatic organisms, but it can lead to a substantial increase in chemical and biological oxygen demands. The contamination of water with EG facilitates the rapid growth of microbial biofilms, which decreases the concentration of dissolved oxygen in water and negatively affects overall biodiversity. The development of a simple method to decompose EG with high efficiency and low operating costs is important. This study revealed that ethylene glycol can be completely oxidized using UV-C activated hydrogen peroxide (H_2_O_2_/UV-C) at a high rate (up to 56 mg L^−1^ h^−1^) at an optimum EG:H_2_O_2_ molar ratio of 1:10–1:15. Air purging the reaction mixture at 1000 cm^3^ min^−1^ increases the EG mineralization rate up to two times because the simultaneous action of UV-activated H_2_O_2_ and O_2_ (H_2_O_2_ + O_2_/UV-C) leads to a synergistic effect, especially at low EG:H_2_O_2_ ratios. The kinetics and mechanism of EG degradation are discussed on the basis of the concentration profiles of ethylene glycol and intermediate products.

## 1. Introduction

Air transport plays an important role in the modern world, providing fast and easy transportation of people, mail, and goods over long distances [1]. Major airports are crucial transport hubs that consume energy and resources but generate many types of emissions, including greenhouse gases, volatile organic compounds (VOCs), organics, and lubricants [2]. Furthermore, aircraft deicing fluids (ADFs) are actively used during the cold season in northern regions to remove ice and snow deposits from aircraft surfaces and prevent their further accumulation. ADFs consist of freezing point depressants (50–75%), water (50–25%), and additives (e.g., colorants, thickeners, corrosion inhibitors, surfactants, and pH modifiers) [3]. ADFs are categorized into four types according to their composition, properties, and application fields [4,5]. Deicing fluids of type I have low viscosity and are predominantly used to remove ice and snow from aircraft surfaces. Type II, III, and IV ADFs have higher viscosities due to the presence of thickening agents, which allow the formation of a thin-film coating on the external surface of aircraft elements to prevent the deposition and accumulation of ice. The main difference between these types of ADFs is attributed to the stability of the formed anti-ice coating at high aircraft speeds. Currently, compositions corresponding to types I and IV ADFs are the most commonly used ADFs at airports [6]. Glycols, namely ethylene glycol (EG) and propylene glycol (PG), serve as depressants in many types of fluids because they have relatively low freezing points. Propylene glycol is mostly employed in the USA and Western Europe, whereas EG-based ADFs are more common in Canada and Russia [7]. The concentration of glycols in the wastewater of large transport hubs during the cold season can significantly exceed threshold limit values, thus leading to environmental pollution [8].

Many studies have shown that ethylene glycol alone has relatively low acute and chronic toxicity to aquatic organisms [9,10,11,12,13]. For example, Pillard [12] reported that the lethal concentration of EG, when 50% of the population dies within 48 h (i.e., 48 h LC_50_), is 82 g L^−1^ for the fathead minnow *Pimephales promelas* and 34 g L^−1^ for the water flea *Ceriodaphnia dubia*. Chronic toxicity can be expressed as the inhibitory concentration when growth is inhibited by 25% of the population (i.e., IC_25_). In the case of *C. dubia*, the IC_25_ of EG was 12 g L^−1^. The acute toxicity of some ADFs can be substantially greater than that of EG per se due to the presence of benzotriazoles, which are used as corrosion inhibitors [14]. More details on the toxicity of EG and EG-based ADFs are summarized in Table 1.

Indirect factors can substantially affect aquatic biota in contaminated water. Microorganisms are able to utilize ethylene glycol as a simple carbon source, which promotes their rapid growth [16,17,18]. The biological oxidation of ethylene glycol during this process consumes a large amount of oxygen and leads to a deficiency of dissolved oxygen in water [18]. This phenomenon can harm fish and macroinvertebrate populations and negatively affect overall biodiversity [6]. Metabolites of ethylene glycol also have toxic effects. For example, oxalic acid can form crystals in fish kidneys, thus leading to tissue necrosis and other forms of damage [10].

Special procedures are employed at airports to prevent the adverse effects of ADFs on the environment. Major airports have special pads for performing anti-icing operations. These pads are connected to a collection system that prevents ADF from discharging into runoff water [19]. Distillation of the collected solution can enable the recycling and reuse of glycols. However, this is an energy-demanded process, which is not economically justified when the glycol concentration is lower than 5% [20,21]. For this reason, other purification methods are preferable for the treatment of low-concentrated solutions to decompose ethylene glycol. Controlled biodegradation in wetlands and soil can be used for the efficient treatment of wastewater [22]. On the other hand, these systems require large land areas and are strongly affected by the ambient temperature and concentration of dissolved oxygen and other contaminants [23]. Open-air systems can also attract undesirable wildlife. Anaerobic oxidation systems do not have these drawbacks and are relatively stable, but they are not able to treat wastewater seasonally containing ethylene glycol at relatively high concentrations [24].

One promising way to solve the problem of water contamination with ethylene glycol is the application of advanced oxidation processes (AOPs). These processes include electrochemical oxidation [25,26], Fenton and photo-Fenton reactions [27,28,29], photocatalysis [30,31,32], ozonation [33,34], and treatment with UV-activated hydrogen peroxide or persulfate [35,36,37]. Hydrogen peroxide is an environmentally friendly reagent that can be used as a strong oxidant in the process of water treatment. H_2_O_2_ can generate hydroxyl radicals OH^•^ with a redox potential of E° = 2.73 V (vs. RHE), which is high enough to oxidize almost all organic compounds [38,39]. However, an important aspect of H_2_O_2_ application is the requirement for its activation to generate a high number of hydroxyl radicals. H_2_O_2_ can be activated sonochemically [40], photochemically [41], or by using Fenton and Fenton-like systems [42]. The activation of H_2_O_2_ with UV-C light is easy to operate and does not require additional electrolytes or catalysts. For these reasons, H_2_O_2_/UV-C oxidation systems are promising for the efficient treatment of water contaminated with ethylene glycol [43,44]. The application of this method in practice requires the determination of the optimal parameters of EG oxidation and the search for approaches to intensify EG oxidation to avoid excessive consumption of oxidizing agents. However, a few studies related to the process of EG oxidation in H_2_O_2_/UV-C systems have been published. For example, McGinnis et al. [29,45] reported the oxidation of EG in H_2_O_2_/UV-C, Fe^2+^/UV-C, and H_2_O_2_/Fe^2+^/UV-C systems. In these studies, a pathway based on a sequence of oxidation steps was proposed for EG degradation. The validity of the proposed EG oxidation pathway in the H_2_O_2_/UV-C system was questioned in [46], where the authors suggested the existence of two parallel pathways of EG oxidation with the formation of oxalic acid and formic acid. Similar pathways were proposed in [35] for EG oxidation using UV-activated persulfate. This work aimed to analyze the kinetics of EG degradation during treatment in the H_2_O_2_/UV-C system. The quantitative characteristics of EG degradation were obtained by analyzing total organic carbon (TOC) and chemical oxygen demand (COD) during irradiation. The effects of the EG:H_2_O_2_ ratio and air purging on the EG degradation rate were studied. Finally, the mechanism of EG degradation is discussed on the basis of the data of the formed intermediates, which are analyzed via HPLC.

## 2. Results and Discussion

### 2.1. Effect of the EG:H_2_O_2_ Molar Ratio

Kinetic experiments on EG degradation were performed in a batch photoreactor with H_2_O_2_ added as a strong oxidant. TOC and COD were monitored during the experiments to analyze the kinetics of EG degradation. A change in the COD allows us to analyze partial oxidation of the initial substrate and intermediates, whereas a change in the TOC reveals complete oxidation of organic matter to CO_2_. The reactor was equipped with an inner-placed UV-C lamp to activate the generation of hydroxyl radicals from H_2_O_2_ and increase the reaction rate (see Section 3 and Appendix A).

Initially, a series of preliminary experiments was performed to confirm the synergistic effect of all the components when they were employed simultaneously. First, the separate effect of H_2_O_2_ on EG oxidation was investigated in a model solution containing EG (500 mg L^−1^ or 8.05 mM) and H_2_O_2_, the initial concentration of which was 10 times greater than that of EG. According to the stoichiometry of complete oxidation:C_2_H_6_O_2_ + 5H_2_O_2_ = 2CO_2_ + 8H_2_O(1)
the molar ratio of reagents in this case (EG:H_2_O_2_ = 1:10) corresponded to the twofold molar excess of H_2_O_2_ toward the stoichiometric value.

Figure 1a shows no evident TOC reduction in the experiment without UV-C irradiation, whereas the COD value slightly decreases over time. A change in the COD allows the conclusion of a partial oxidation of EG with H_2_O_2_, but the corresponding COD rate is extremely low (11 mg L^−1^ h^−1^). This confirms that H_2_O_2_ by itself poorly oxidizes organics, even low-molecular-weight compounds such as ethylene glycol and that the activation of H_2_O_2_ is essential for efficient EG degradation.

It is well known that UV-C light causes photolysis of organic compounds and results in the degradation of organic contaminants in water [47]. For this reason, the separate effect of UV-C light was also studied. Figure 1b shows no evident TOC reduction during irradiation of the EG solution (500 mg L^−1^) with UV-C light, which confirms that no complete oxidation of ethylene glycol occurred. At the same time, the COD monotonically decreases with a rate of 36 mg L^−1^ h^−1^, which indicates a partial oxidation of EG due to the impact of UV-C light and ozone photogenerated from dissolved oxygen in water. Glycol aldehyde, acetaldehyde, formaldehyde, and formic acid were identified as the main oxidation products formed under these conditions. Appendix A shows the concentration profiles of EG and formed products to illustrate changes in the composition of the reaction medium during UV-C irradiation of the EG solution. Notably, the COD rate under UV-C light is three times greater compared to the case of the treatment with H_2_O_2_ only, but it also has a low absolute value, which is not high enough for practice. As will be shown later, the combined action of H_2_O_2_ and UV-C light allows a sharp increase in the rate of EG degradation.

The irradiation of H_2_O_2_ with UV-C light leads to the formation of OH^•^ radicals, which are reactive oxygen species [48]. This process is commonly referred to as the activation of hydrogen peroxide [49]. A high number of formed OH^•^ radicals promotes the generation of many organic radicals due to their interactions with organic compounds (Appendix A). Figure 2 shows the TOC and COD plots during the experiment on EG degradation with the H_2_O_2_/UV-C system when the initial molar ratio of EG:H_2_O_2_ was adjusted to 1:25.

Both the TOC and COD values monotonically decrease during irradiation. An induction period is present in the TOC plot, whereas the COD starts to decrease immediately after turning the UV-C light on. This means that EG oxidation occurs through the formation of intermediates, but under irradiation, these intermediates are completely oxidized with the formation of CO_2_ as the final product, which is released to the gas phase. The TOC plot during the whole period of irradiation until the complete removal of organic matter poorly corresponds to the kinetics of a first-order reaction due to the multistage nature of the oxidation process. To compare the efficiency of EG degradation in different experiments, the TOC values after the induction period were linearly approximated, and the estimated rate of TOC removal was used as an efficient reaction rate. When the EG:H_2_O_2_ ratio was 1:25, the rate of TOC removal was 47 mg L^−1^ h^−1^ (Figure 2a). Notably, the COD data can also be used for this purpose because the relative reduction in COD is the same as that in TOC (i.e., 24% h^−1^, as shown in Figure 2) despite different absolute values.

An important aspect of reaction engineering is the optimization of operating parameters to maximize the efficiency of oxidant utilization [50]. The effect of the EG:H_2_O_2_ molar ratio on the kinetics of EG mineralization was investigated on the basis of the TOC removal rate. As mentioned above, the stoichiometric ratio for the complete oxidation of EG with H_2_O_2_ is 1:5 (see Equation (1)). The results of the experiments revealed that the H_2_O_2_ amount at the stoichiometric value or lower is not high enough for efficient degradation of EG, and an excess of H_2_O_2_ is needed. Figure 3 shows that the rate of EG mineralization has a maximum of 56 mg L^−1^ h^−1^ at an EG:H_2_O_2_ ratio of 1:10–1:15. This value corresponds to the two- and threefold molar excesses of H_2_O_2_ toward the stoichiometric value, respectively.

However, a further increase in the concentration of hydrogen peroxide above these values does not increase the rate of EG mineralization but rather reduces it. At high H_2_O_2_ concentrations, OH^•^ radicals interact with molecules of hydrogen peroxide to form hydroperoxyl radicals (HO_2_^•^) via the following reaction [51]:H_2_O_2_ + OH^•^ = H_2_O + HO_2_^•^(2)

The formed HO_2_^•^ radicals are less active than OH^•^ radicals, which leads to a decrease in the overall reaction rate.

Thus, fast EG mineralization in the H_2_O_2_/UV-C system can be achieved when the H_2_O_2_ concentration is 2–3 times greater than the stoichiometric value.

### 2.2. Intensification of the Oxidation Process

Several approaches were tested to intensify EG degradation in the H_2_O_2_/UV-C system and reduce operational costs via a shorter irradiation time and a lower amount of added H_2_O_2_. Purging the reaction medium with oxygen at a volume flow rate of 200 cm^3^ min^−1^ resulted in a substantial increase in the rate of EG mineralization. Figure 4 shows that the TOC removal rate in the H_2_O_2_ + O_2_/UV-C system is up to two times greater than that in the experiments without oxygen purging (i.e., H_2_O_2_/UV-C). In the case of the stoichiometric mixture (EG:H_2_O_2_ = 1:5, Figure 4a), 90% TOC removal under oxygen purging occurred after 3.75 h, whereas 5 h was required for the same depth of mineralization in the H_2_O_2_/UV-C system. This occurred because oxygen purging increased the initial rate of EG mineralization by 1.4 times from 46 to 64 mg L^−1^ h^−1^. Furthermore, oxygen purging allows efficient EG degradation at EG:H_2_O_2_ molar ratios lower than the stoichiometric value. Figure 4b shows that the TOC removal rate in the H_2_O_2_ + O_2_/UV-C system with an EG:H_2_O_2_ ratio of 1:3 reaches 56 mg L^−1^ h^−1^, which is the same TOC rate as the maximum in the H_2_O_2_/UV-C system observed at an EG:H_2_O_2_ ratio of 1:10–1:15 (see Figure 3). This means that oxygen can serve as an efficient oxidant under the conditions of this experiment, thus resulting in intensification of the oxidation process.

Other experiments with air purging at flow rates of 200 and 1000 cm^3^ min^−1^ were carried out to estimate the main reason for the observed effect on the increase in the EG oxidation rate. Figure 5 shows that the rates of EG mineralization in the experiments with an airflow rate of 1000 cm^3^ min^−1^ and an oxygen flow rate of 200 cm^3^ min^−1^ are similar, which is probably due to the same content of oxygen in both gas flows. Otherwise, purging the reaction medium with air at a flow rate of 200 cm^3^ min^−1^ led to a lower increase in the TOC removal rate than that in the other cases, probably due to the lower content of oxygen in this gas flow. This finding suggests that oxygen plays a crucial role in the enhancement of EG mineralization and that oxygen or air purging leads to an increase in the reaction rate via an increase in the concentration of dissolved oxygen in the reaction solution.

The results discussed above reveal the synergistic effect between UV-activated H_2_O_2_ and O_2_, the simultaneous action of which substantially enhances EG mineralization, especially at a low EG:H_2_O_2_ molar ratio. Air purging of the reaction medium at a sufficient flow rate may reduce the required amount of H_2_O_2_ by 2–3 times while maintaining a high rate of EG degradation and depth of its mineralization.

### 2.3. Pathways of EG Degradation

The oxidation of ethylene glycol predominantly occurs in a stepwise manner with the formation of many organic intermediates, including glycol aldehyde, formaldehyde, glyoxal, glycolic acid, glyoxylic acid, oxalic acid, and formic acid [29,33,52,53,54]. To investigate the reasons for the strong difference in the mineralization rates under the EG treatment in the H_2_O_2_/UV-C and H_2_O_2_ + air/UV-C systems, the concentrations of the abovementioned reaction components were monitored via HPLC analysis (Appendix A). Importantly, other complex products may also form in radical processes due to dehydration, esterification, or polymerization reactions [29,35].

Figure 6 shows the concentration profiles of EG and the main intermediates detected in the H_2_O_2_/UV-C system. The concentration of ethylene glycol starts to decrease rapidly after the UV-C light is turned on and reaches zero after the long-term irradiation (Figure 6a). Its concentration profile corresponds well to the kinetics of a first-order reaction with an efficient rate constant (k) of 3.5 ± 0.1 h^−1^ (see Appendix A). Simultaneously, formic acid forms the main oxidation product during the initial period. Its concentration increases to 1.76 mM for the first 0.5 h and further decreases gradually (Figure 6a). Thus, C–C bond cleavage can be considered a crucial step of the oxidation process at high concentrations of H_2_O_2_.

Among the aldehydes, glycol aldehyde had the highest concentration (1.3 mM) after 0.5 h of irradiation (Figure 6b). The maximum concentration of formaldehyde was three times lower than that of glycol aldehyde. The formation of formaldehyde may occur via the oxidation of glyoxal, and it is consequently oxidized to formic acid. The concentration of glycolic acid reached a maximum after 1.5 h of irradiation and then decreased with a simultaneous increase in the concentration of glyoxylic and oxalic acids because of partial oxidation (Figure 6c). Notably, the concentration of glyoxylic acid reached the maximum value after 4 h of irradiation and did not further decrease because of the consumption of the major H_2_O_2_ portion at this time (Figure 7d).

The presence of glyoxal and the increase in formic acid concentration in the reaction medium were not observed after 2 h of irradiation, but changes in the concentrations of glycolaldehyde, glycolic acid, glyoxylic acid, and oxalic acid continued. From these results, it can be concluded that the oxidation of glycolic acid likely proceeded, with the formation of glyoxylic and oxalic acids only. The results obtained confirm previously observed effects, such as the absence of formic acid formation long after the beginning of irradiation during the EG oxidation process in the H_2_O_2_/UV-C system [46]. Similar to the results above, the formation of oxalic acid was observed after a long irradiation time, in contrast to formic acid; therefore, the oxidation of EG includes two pathways involving the formation of formic and oxalic acids.

In the process of EG oxidation, the first step can be expected as an attack of EG by OH^•^ radical with the formation of HO–^•^CH–CH_2_–OH and, subsequently, glycol aldehyde (Appendix A). However, it is difficult to predict a detailed mechanism of further radical transformations due to the presence of various reactive oxygen species [55,56]. The main pathways of the EG oxidation process can be proposed on the basis of the concentration profiles of the oxidation intermediates. The results of the HPLC analysis allow us to propose two main pathways for EG degradation (Figure 6d). The first involves C–C bond cleavage in ethylene glycol through glycolaldehyde and glyoxal, resulting in the formation of formic acid. As discussed in [55,56,57,58], glyoxal formed as a product of EG degradation can be easily oxidized by hydrogen peroxide even without UV-C irradiation to formic acid, thus resulting in a high concentration of HCOOH for the initial period of irradiation. This sequence is proposed on the basis of the observed sharp increase in the glycolaldehyde and formic acid concentrations, coupled with a slight increase in the glyoxal concentration during the first 0.5 h of irradiation. No significant differences in the concentrations of glyoxylic acid or oxalic acid were observed during the first 0.5 h of irradiation. This route clearly plays a main role during the first minutes of the process. The second pathway (the C_2_-pathway) is the sequential oxidation of ethylene glycol to oxalic acid. Both oxalic acid and formic acid can be easily oxidized to CO_2_ under reaction conditions [29].

As discussed above, air purging leads to a substantially higher rate of TOC removal. Figure 7 shows the concentration profiles of EG and the main intermediates in the case of the H_2_O_2_ + air/UV-C system. The general trends are similar to those of the H_2_O_2_/UV-C system: fast decrease in the EG concentration and high concentration of formic acid in the first minutes of irradiation (Figure 7a), relatively high concentration of glycol aldehyde (Figure 7b), and partial oxidation of glycolic acid to glyoxylic and oxalic acids (Figure 7c).

On the other hand, the concentration of formic acid in the H_2_O_2_ + air/UV-C system (4.13 mM) was approximately two times greater than that in the H_2_O_2_/UV-C system, whereas the concentration of glycol aldehyde was lower and had a substantially higher removal rate (Figure 7b). The concentrations of glyoxylic acid and oxalic acid continued to decrease after they reached their maximum values. The concentrations of glyoxylic and oxalic acids with the highest values were 0.84 and 0.90 mM, respectively. This value is significantly greater than that of the H_2_O_2_/UV-C system (0.56 mM for glyoxylic acid and 0.19 mM for oxalic acid), which indicates a higher rate of EG oxidation via the C_2_-pathway under air purging.

To determine the difference in the efficiency of H_2_O_2_ utilization, the concentration of hydrogen peroxide was monitored during the reaction. The concentration profiles of H_2_O_2_ substantially differ in the experiments with and without air purging (Figure 7d). In the H_2_O_2_/UV-C system, the H_2_O_2_ concentration profile corresponds to the kinetics of a first-order reaction. In contrast, the H_2_O_2_ concentration decreases more slowly in the presence of air purging, and its kinetic plot has another form. This fact indicates that O_2_ plays the role of an oxidant, reducing the amount of hydrogen peroxide required for the complete mineralization of EG. According to Vel Leitner and Dorè [54], the process of glycolic acid oxidation to oxalic acid in the H_2_O_2_/UV-C system can be increased by the addition of oxygen, which reacts with organic radicals. This leads to faster oxidation of glycol aldehyde to oxalic acid via the C_2_-pathway in the H_2_O_2_ + air/UV-C system. Similarly, it can be proposed that the process of ethylene glycol oxidation to glyoxal is also affected by oxygen (see Appendix A), thus resulting in a higher concentration of formic acid during the initial period of irradiation via the C–C bond cleavage pathway. On the other hand, the rate of formic acid oxidation in the H_2_O_2_/UV-C system can be suppressed by oxygen, as shown by Aristova et al. [59]. The oxidation of formic acid is the last step in EG degradation via the C–C cleavage pathway; thus, TOC removal in the H_2_O_2_ + air/UV-C system can be limited by the oxidation of formic acid despite the increased oxidation of other organic intermediates. After a substantial decrease in the concentration of formic acid, the oxidation of C_2_-acids becomes the second limiting factor in the process of EG mineralization.

Furthermore, different photochemical reactions, such as polymerization, condensation, and esterification, can lead to the formation of difficult-to-oxidize compounds (C_3_ and C_4_ organic molecules, for instance, malonic and succinic acids) [55,57,60]. These compounds can prevent fast removal of TOC and reduce the efficiency of H_2_O_2_ utilization. An acceleration of the oxidation process due to an increased concentration of oxygen dissolved in the reaction solution may help reduce the contribution of the photochemical reactions mentioned above because oxygen reacts with intermediate organic radicals.

## 3. Materials and Methods

Ethylene glycol (EG) of analytical grade from Reachim (Moscow, Russia) was mixed with deionized water (18.2 MΩ cm) to prepare model reaction solutions with an EG concentration of 500 mg L^−1^. The experiments on EG degradation were performed using these solutions in addition to hydrogen peroxide as an oxidizing agent. The molar ratio between EG and H_2_O_2_ was varied over a wide range from 1:0 to 1:100. A commercially available 30% H_2_O_2_ solution of laboratory grade (Lega, Moscow, Russia) was titrated with a standardized solution of KMnO_4_ to determine the exact concentration of H_2_O_2_. Then, a certain aliquot of H_2_O_2_ was added to the reaction mixture to achieve the required molar ratio toward EG.

Kinetic experiments of EG degradation using H_2_O_2_ under activation with UV-C light were carried out in a 1.5 L glass vessel with an inner quartz tube (Figure 8a). The exterior of the experimental setup is shown in Appendix A. A GPH212T5VH/4 germicidal lamp (Heraeus Group, Hanau, Germany) with an electrical power of 10 W was used as a source of UV-C light. Figure 8b shows the emission spectrum of the lamp used, which was measured outside the quartz tube via an ILT950 spectroradiometer (International Light Technologies, Peabody, MA, USA). Importantly, the 185 nm line is not represented in the spectrum shown in Figure 8b due to the technical limitations of the spectroradiometer used, but according to the manufacturer’s data, its intensity is 5–6 times lower than that of the 254 nm line. In routine experiments, 1 L of EG solution with added H_2_O_2_ was placed in the reaction vessel and magnetically stirred for 5 min before the UV-C source was turned on. The moment at which the UV-C light was turned on was considered the start of the reaction. The initial temperature of the reaction medium was 22 ± 1 °C (see Appendix A). Blank experiments without H_2_O_2_ or UV-C light were also performed to determine the synergistic effect of both factors. In several experiments focused on the effect of oxidizing agents, the reaction mixture was continuously purged with oxygen or air at a controlled flow rate of 200–1000 cm^3^ min^−1^.

EG degradation was studied by monitoring a combination of parameters, namely total organic carbon (TOC) and chemical oxygen demand (COD), in probes sampled periodically from the reaction mixture during the experiment. The total content of organic carbon was measured via a TOC N/C multi 3100 total organic carbon analyzer (Analytik Jena GmbH, Jena, Germany). For routine measurements, 10 mL of the reaction medium was pre-acidified with 100 μL of 2 M HCl (Reachim, Moscow, Russia), placed in an autosampler of a TOC N/C multi 3100 analyzer, and purged with oxygen under stirring to remove dissolved CO_2_. After that, three aliquots of 500 μL were taken automatically to measure the average value of the TOC (mg L^−1^). The COD (mg L^−1^) was measured according to the ISO 6060:1989 standard [61], but special pretreatment was performed to eliminate the adverse effect of H_2_O_2_. A total of 10 mL of the reaction medium was mixed with 100 mg of MnO_2_ (Reachim, Moscow, Russia) and magnetically stirred overnight. The mixture was subsequently filtered through a 0.22 μm PTFE membrane. The complete removal of H_2_O_2_ was confirmed by measuring the H_2_O_2_ concentration via a colorimetric enzymatic method, also known as the Trinder method. First, 1 mL of commercial GOD-PAP glucose oxidase enzymatic reagent (Vector Best, Novosibirsk, Russia) and 2 mL of phosphate buffer with a pH of 7.0 were added to 1 mL of the sample mixture, followed by storage at 37 °C for 10 min. In the presence of H_2_O_2,_ horseradish peroxidase-assisted co-oxidation of phenol and 4-aminoantipiryne by H_2_O_2_ led to the formation of quinoneimine dye, which can be identified optically with a maximum absorbance at 510 nm (Figure 9a). No H_2_O_2_ was detected in the solutions after the proposed pretreatment, which allowed the correct measurement of the COD. For the routine procedure of COD measurement, 2 mL of pretreated reaction medium was mixed with 0.5 mL of 0.5 M K_2_Cr_2_O_7_ solution in deionized water and 2.5 mL of 0.04 M Ag_2_SO_4_ in sulfuric acid (Reachim, Moscow, Russia). The mixture was heated under reflux in a sand bath at 150 °C for 2 h. After heating, the solution was cooled naturally and transferred to a quartz cuvette with an optical path length of 1 cm. Absorption spectra were recorded on a Cary 300 spectrophotometer (Agilent, Santa Clara, CA, USA) using a similar solution with deionized water as a reference. The absorbance at 600 nm attributed to light absorption by Cr^3+^ ions was used for the determination of the COD value on the basis of a calibration plot performed with potassium hydrogen phthalate (Sigma-Aldrich, St. Louis, MA, USA). Notably, the COD values measured for the samples pretreated with MnO_2_ to remove H_2_O_2_ residue (630 ± 6 mg L^−1^) corresponded well with the theoretical values regardless of the amount of H_2_O_2_ added over the whole studied range (Figure 9b).

The concentrations of EG and the main oxidation intermediates were analyzed via high-performance liquid chromatography (HPLC) on an Acquity H-Class system (Waters Corporation, Milford, MA, USA) equipped with a refractive index detector (RID) and photodiode array detector (PDA). The HPLC system was also equipped with an ion-exclusion Repromer H column of 250 × 4.6 mm filled with polymeric sorbent granules of 9 μm (Dr. Maisch HPLC GmbH, Ammerbuch, Germany). A 5 mM solution of sulfuric acid in deionized water was used as the eluent, which was pumped at a flow rate of 0.5 cm^3^ min^−1^. The RID and PDA detectors were connected sequentially, and the PDA wavelength was set at 210 nm. The temperature of the RID cell and column thermostat were set at 50 °C, whereas the autosampler temperature was maintained at 25 °C. The sample volume was 10 μL. The RID and PDA signals were analyzed via Empower 3 software (Waters Corporation). The retention times of the degradation products and examples of the experimental chromatograms can be found in the Appendix A.

## 4. Conclusions

In this study, the kinetics of ethylene glycol degradation in aqueous solution upon the addition of hydrogen peroxide was investigated under irradiation with UV-C light. The highest TOC removal rate (56 mg L^−1^ h^−1^) in the H_2_O_2_/UV-C system was observed at EG:H_2_O_2_ molar ratios of 1:10 and 1:15, which corresponded to two- and threefold molar excesses of H_2_O_2_ toward the stoichiometric value, respectively. A further increase in the TOC removal rate was achieved by purging the reaction solution with oxygen or air. The simultaneous action of UV-C-activated H_2_O_2_ and O_2_ leads to a synergistic effect on the enhancement of EG mineralization, especially at low EG:H_2_O_2_ molar ratios. The TOC removal rate in the H_2_O_2_ + O_2_/UV-C system was up to two times greater than that in the experiments without oxygen (i.e., H_2_O_2_/UV-C).

Two main pathways of EG mineralization are proposed based on the results of the HPLC analysis. The first leads to the formation of formic acid via C–C bond cleavage as the main oxidation product and plays a crucial role in EG oxidation during the initial period of irradiation. The second is a pathway of sequential oxidation of EG to oxalic acid. This pathway dominates at low concentrations of H_2_O_2_. An increase in the concentration of dissolved oxygen leads to an increase in the oxidation rate via both proposed pathways. Oxygen acts as an oxidant, thus reducing the amount of hydrogen peroxide required for the complete mineralization of EG. The kinetic aspects disclosed in this study reveal the importance of combining many components in AOPs to reduce the required amounts of valuable oxidants, accelerate the mineralization process, and reduce operating costs by achieving synergistic effects. This study can be the basis for further research on the composition effect of the treated medium (pH, salt ions, and combination of pollutants) on the degradation efficiency in designing high-performance water treatment systems.

## Figures and Tables

**Figure 1 molecules-30-00049-f001:**
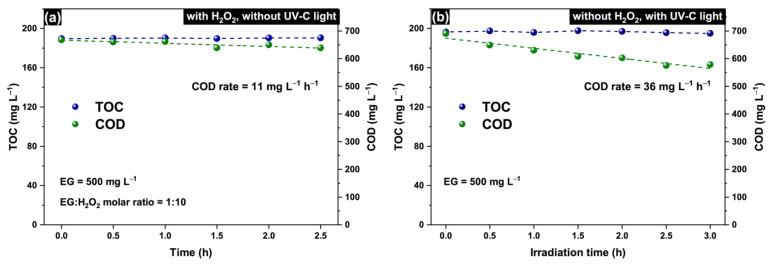
Time dependence of the TOC and COD in EG solution treated with (**a**) H_2_O_2_ (without irradiation) or (**b**) UV-C light (without the addition of H_2_O_2_).

**Figure 2 molecules-30-00049-f002:**
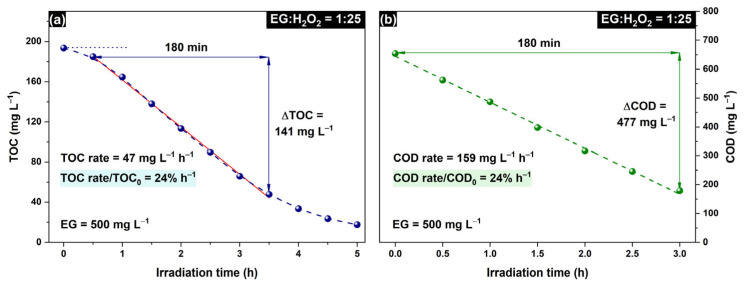
Time dependence of (**a**) TOC and (**b**) COD during irradiation of EG solution (500 mg L^−1^) containing H_2_O_2_ (EG:H_2_O_2_ = 1:25) with UV-C light. Red line in (**a**) shows a linear approximation of the experimental data.

**Figure 3 molecules-30-00049-f003:**
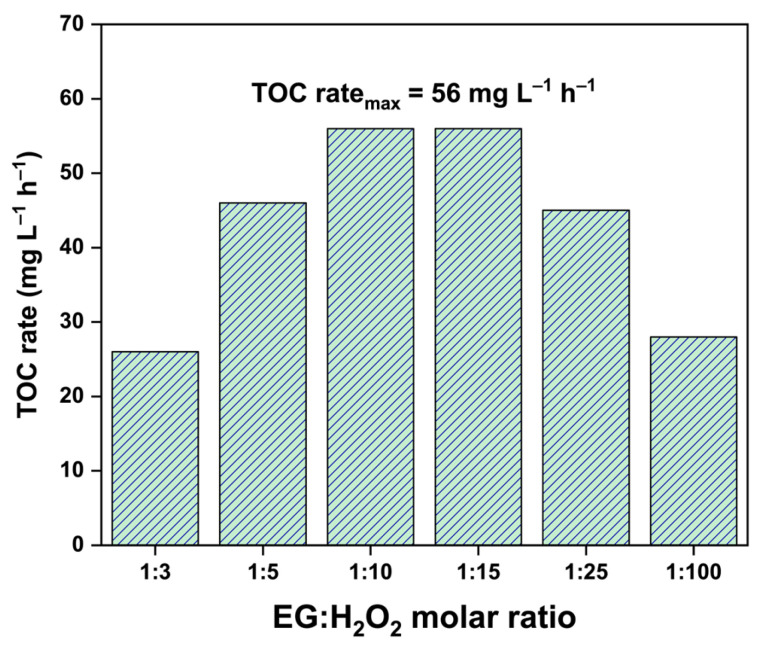
Effect of the EG:H_2_O_2_ molar ratio on the TOC removal rate during EG degradation (initial EG concentration is 500 mg L^−1^, initial temperature is 22 ± 1 °C).

**Figure 4 molecules-30-00049-f004:**
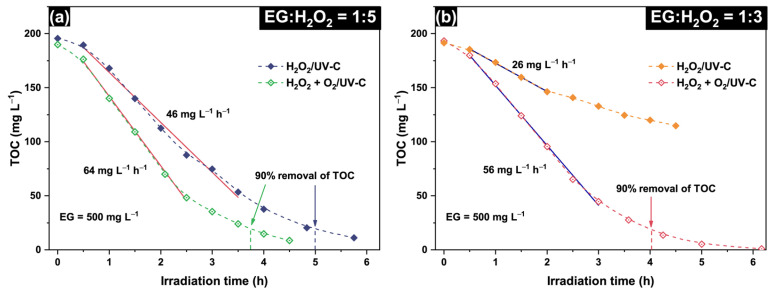
Comparison of EG mineralization in the H_2_O_2_/UV-C and H_2_O_2_ + O_2_/UV-C systems during the experiments with EG:H_2_O_2_ molar ratios of (**a**) 1:5 and (**b**) 1:3 (initial EG concentration is 500 mg L^−1^, initial temperature is 22 ± 1 °C). Red lines in (**a**) and blue lines in (**b**) show linear approximations of the experimental data.

**Figure 5 molecules-30-00049-f005:**
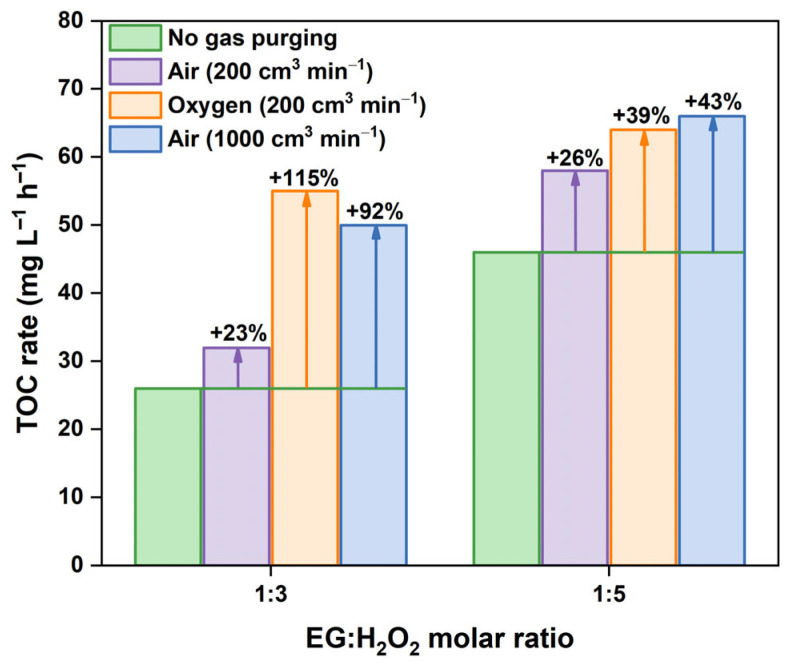
Comparison of EG mineralization rates under different purging conditions of the reaction medium (initial EG concentration is 500 mg L^−1^, initial temperature is 22 ± 1 °C).

**Figure 6 molecules-30-00049-f006:**
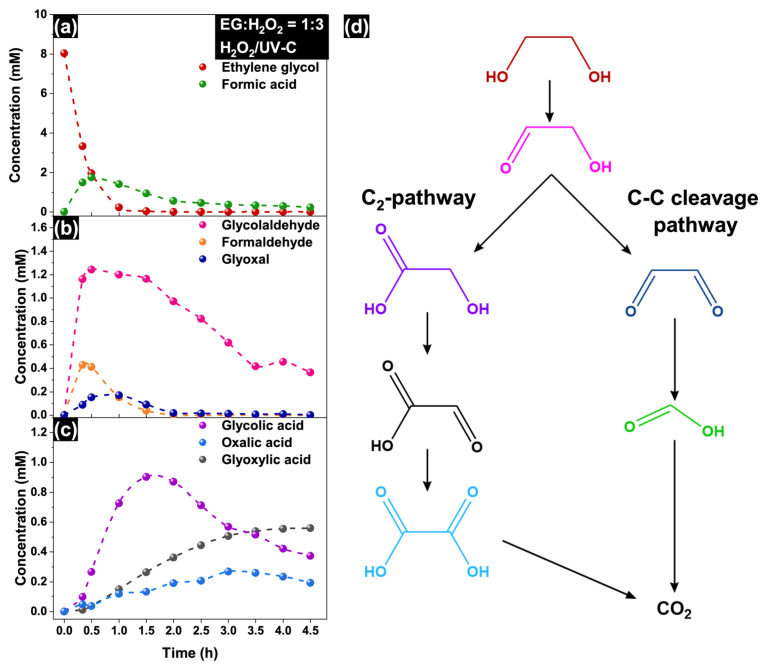
Concentration profiles of (**a**) EG and formic acid, (**b**) aldehydes, and (**c**) C_2_-carboxylic acids during EG treatment in the H_2_O_2_/UV-C system (EG:H_2_O_2_ is 1:3, initial EG concentration is 500 mg L^−1^, initial temperature is 22 ± 1 °C); (**d**) main pathways of EG degradation.

**Figure 7 molecules-30-00049-f007:**
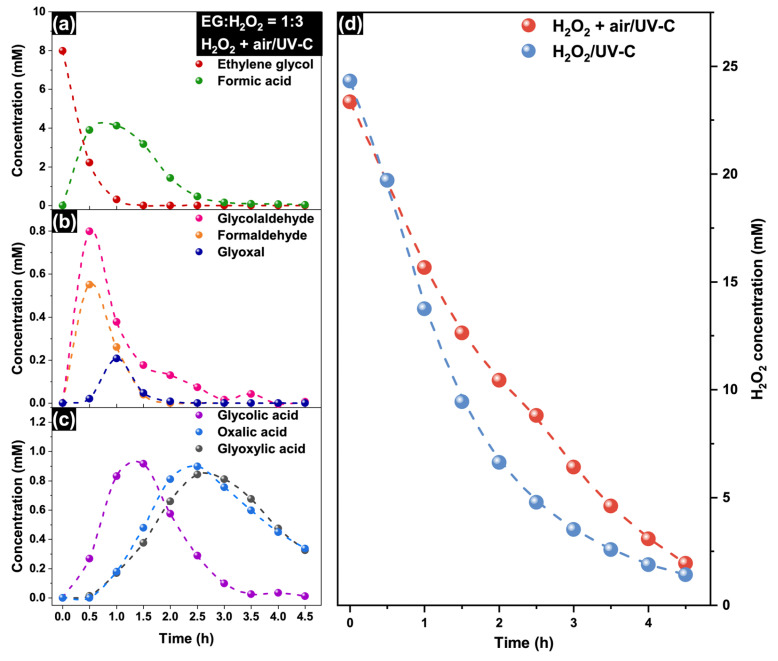
Concentration profiles of (**a**) EG and formic acid, (**b**) aldehydes, and (**c**) C_2_-carboxylic acids during EG treatment in the H_2_O_2_ + air/UV-C system (EG:H_2_O_2_ is 1:3, airflow is 1000 cm^3^ min^−1^, initial EG concentration is 500 mg L^−1^, initial temperature is 22 ± 1 °C); (**d**) comparison of H_2_O_2_ concentration profiles in the H_2_O_2_/UV-C and H_2_O_2_ + air/UV-C systems.

**Figure 8 molecules-30-00049-f008:**
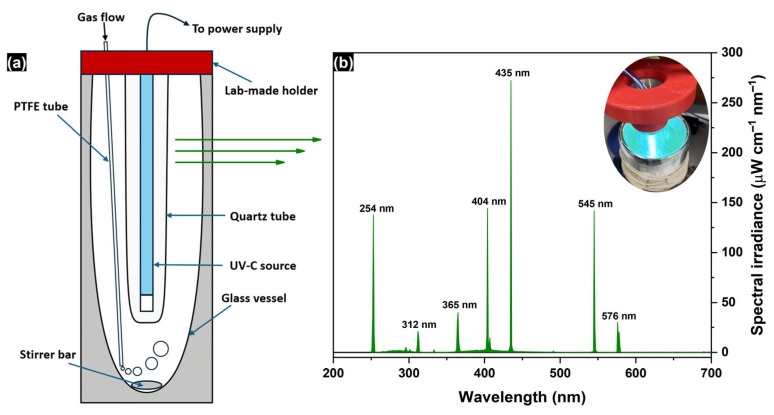
(**a**) Scheme of the reaction vessel (blue arrows show the vessel components, green arrows illustrate the emission from the UV-C lamp); (**b**) emission spectrum of the used germicidal UV-C lamp in the region of 200–700 nm (the inset shows a photograph of the lamp installed into the vessel).

**Figure 9 molecules-30-00049-f009:**
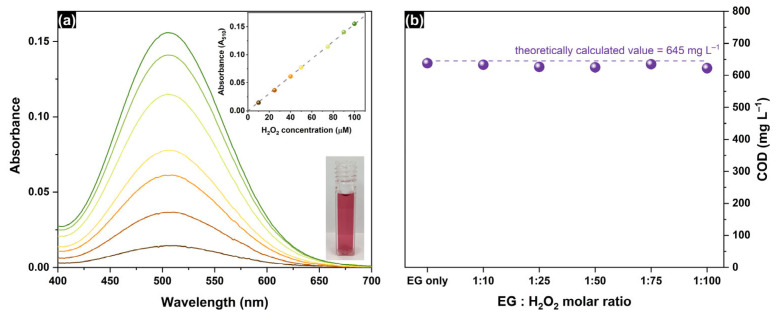
(**a**) Absorbance spectra of the quinoneimine dye formed in the Trinder method (different colors correspond to different concentrations of H_2_O_2_, the inset shows the calibration plot for the determination of H_2_O_2_ concentration); (**b**) COD values measured for MnO_2_-pretreated EG solutions (500 mg L^−1^) containing different amounts of added H_2_O_2_.

**Table 1 molecules-30-00049-t001:** Toxicity of EG and EG-based ADFs.

Reagent	Model Organism	Indicator	Concentration (mg L^−1^)	Ref.
Ethylene glycol	*Ceriodaphnia dubia*	48 h LC_50_	34,440	[12]
		IC_25_	12,310	
Type I ADF		48 h LC_50_	13,140	
		IC_25_	3960	
Ethylene glycol	*Pimephales promelas*	48 h LC_50_	81,950	
		IC_25_	22,520	
Type I ADF		48 h LC_50_	8540	
		IC_25_	3660	
Type I ADF	*Ceriodaphnia dubia*	48 h LC_50_	15,700	[14]
		IC_25_	5470	
Type IV ADF		48 h LC_50_	449	
		IC_25_	113	
Type I ADF	*Pimephales promelas*	96 h LC_50_	24,700	
		IC_25_	4430	
Type IV ADF		96 h LC_50_	371	
		IC_25_	179	
Ethylene glycol	*Selenastrum capricornutum*	IC_25_	5340	[15]
Type I ADF			7610	
Ethylene glycol	*Trachinotus carolinus*	24 h LC_50_	>60,000	[11]

## Data Availability

Data are contained within the article or Appendix A.

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
