# Peer review of "Kinetic Aspects of Ethylene Glycol Degradation Using UV-C Activated Hydrogen Peroxide (H2O2/UV-C)"

_molecules, 2024, doi:10.3390/molecules30010049_

Round 1
Reviewer 1 Report
Comments and Suggestions for Authors
Please see attached file with comments

Please see attached file
Author Response
Response to the Reviewer #1:
Dear the Reviewer #1,
thank you very much for your positive feedback and valuable comments on our work. The manuscript was revised according to your comments and recommendations. Please find the detailed answers and made changes below.
Comment: […] Finally, the subject matter of the MS is well aligned with the scope of the journal Environments, also from MDPI where the audience will most likely offer a wider appreciation of the contribution to the co-authors. Much of the research reported in the MS is conducted at the microscopic and macroscopic scales of the continuum engineering assumption and very litter from the molecular scale.
Response: We really appreciate your intention to attract more attention and enhance the readerships by sending the manuscript to the Environments journal. However, we call your attention to the fact that the Special issue "Advanced Oxidation Processes (AOPs) in Treating Organic Pollutants" of the Molecules journal, to which this manuscript was submitted, is aimed to the advanced oxidation processes, such as Fenton's reaction, photocatalysis, and ozonation, as well as the use of advanced nanomaterials. One of the key points of this special issue is optimization strategies to enhance the efficiency and selectivity of AOPs, to which we believe our manuscript belongs. Thus, we prefer not to change the journal for the submission.
Comment: The Organization of the MS should be adjusted as follows:
1. Introduction
This section seems well written and it seems appropriate for the MS
2. Materials and Methods
The entire current Section 3 should be here, and all or part of current Section 2 should be integrated into this new section. Except the results that should be part of the section suggested below:
3. Research Strategy and Results Obtained
Here the co-authors should describe the overall strategy for the investigation and present the results obtained. The first part should be focused more on the parametric studies and the results obtained. The second part should be the presentation of the role of the degradation mechanism and the results obtained. The description here should be limited to an accurate description of what was observed.
Explanations and supporting evidence of the behaviors should be part of the following section.
4. Discussion of Results
In this section by using facts known from the literature and others obtained in the experiments, the co- authors should offer a compelling discussion explaining and justifying the trends in the results.
5. Summary and Concluding Remarks
In this section, there is an opportunity to summarize key observations and identify limitations of the study. For example, I could not identify a parametric study regarding the pH and this is a factor that has been found to be key in the application of AOP.
Response: We appreciate your intention to improve the reading experience of the submitted manuscript. However, we call your attention to the fact that for manuscript preparation all the authors should be guided by the Instructions for Authors provided by the Molecules journal (https://www.mdpi.com/journal/molecules/instructions). The structure of our manuscript and sequence of sections and their contents completely correspond to these instructions that is confirmed by journal’s staff during technical check.
We agree that other factors (e.g., pH, presence of different ions, ionic strength, combination of contaminants) can affect the efficiency of EG degradation, but their investigation is out of the scopes of this study and may be the object of our future work. A study of the pH effect is mainly interesting from the research point of view because pH changes in large volumes of runoff water (in which it rarely deviates from 6-8 [R1]) do not seem reasonable. In the future, it is possible to study the effect of additional components of real water samples on the process of ethylene glycol oxidation.
It is worth noting that not so many studies on ethylene glycol oxidation in H2O2/UV-C system have been published. This system is commonly used as a model system to determine the main intermediates of EG oxidation [R2, R3] or to compare with S2O82-/UV C system [R4]. In summary, the previously published studies have shown the possibility of using this system for complete mineralization of ethylene glycol and suggested possible routes of the process, although differing depending on each specific study. However, the optimal parameters on H2O2 amount as the oxidizing agent, the kinetic plots of intermediates, and possible approaches to intensify the oxidation process to reduce operating costs were not studied, whereas these aspects are very important for broader practical application of this technique. Therefore, we believe that new data and results of this study can be interesting and useful for readers of the Molecules journal to discover the mechanism of EG degradation and to make efficient H2O2/UV-C oxidation systems for wastewater treatment.
References:
R1. Sulej-Suchomska, A.; Polkowska, Å».; NamieÅ›nik, J. Contaminants in Airport Runoff Water in the Vicinities of Two International Airports in Poland. Pol. J. Environ. Stud. 2012, 21, 725–739.
R2. McGinnis, B.D.; Adams, V.D.; Middlebrooks, E.J. Degradation of Ethylene Glycol in Photo Fenton Systems. Water Res. 2000, 34, 2346–2354, doi:10.1016/S0043-1354(99)00387-5.
R3. Dietrick McGinnis, B.; Dean Adams, V.; Joe Middlebrooks, E. Degradation of Ethylene Glycol Using Fenton’s Reagent and UV. Chemosphere 2001, 45, 101–108, doi:10.1016/S0045-6535(00)00597-X.
R4. Priyadarshini, M.; Ahmad, A.; Das, I.; Ghangrekar, M.M.; Dutta, B.K. Efficacious Degradation of Ethylene Glycol by Ultraviolet Activated Persulphate: Reaction Kinetics, Transformation Mechanisms, Energy Demand, and Toxicity Assessment. Environ. Sci. Pollut. Res. 2023, 30, 85071–85086, doi:10.1007/s11356-023-27596-9.
According to your suggestion and suggestions from the reviewer #2, we have added more contents and deeper discussion of the results and future prospects to the corresponding sections of manuscript. Please find the revised version of manuscript.
Comment: a. The term UV-C can easily be confused with the combination of the use of UV-light with, for example activated carbon. I would suggest using UV-light assisted degradation with the addition of H2O2 to avoid confusion.
Response: We appreciate your intention to make paper clearer for understanding. According to the IUPAC Gold Book [R1] and the Glossary of terms used in photochemistry, 3rd edition (IUPAC Recommendations 2006) [R2], ultraviolet is divided into four sub-bands: VUV, UV C, UV B, and UV A, each within corresponding range of wavelengths. We exclusively use the UV C naming because, as has shown in [R3], only UV C and VUV radiations cause an intensive photolysis of H2O2. From this point of view, use of UV term without specific definition of the main wavelength range affected could lead to a confusion. On the other hand, no articles devoted to UV and activated carbon system, which used such naming, were found by the authors. It is worth noting that activated carbon is commonly shortened to AC in many studies [R4–R6].
Thus, we insist on using this UV C term because it corresponds to the IUPAC recommendations and is unlikely to result in misreading.
References:
R1. International Union of Pure and Applied Chemistry (IUPAC) Ultraviolet., doi:10.1351/goldbook.UT07492.
R2. Braslavsky, S.E. Glossary of Terms Used in Photochemistry, 3rd Edition (IUPAC Recommendations 2006). Pure Appl. Chem. 2007, 79, 293–465, doi:10.1351/pac200779030293.
R3. Goldstein, S.; Aschengrau, D.; Diamant, Y.; Rabani, J. Photolysis of Aqueous H2O2: Quantum Yield and Applications for Polychromatic UV Actinometry in Photoreactors. Environ. Sci. Technol. 2007, 41, 7486–7490, doi:10.1021/es071379t.
R4. Gao, Y.; Yue, Q.; Gao, B.; Li, A. Insight into Activated Carbon from Different Kinds of Chemical Activating Agents: A Review. Sci. Total Environ. 2020, 746, 141094, doi:10.1016/j.scitotenv.2020.141094.
R5. Santoro, C.; Artyushkova, K.; Babanova, S.; Atanassov, P.; Ieropoulos, I.; Grattieri, M.; Cristiani, P.; Trasatti, S.; Li, B.; Schuler, A.J. Parameters Characterization and Optimization of Activated Carbon (AC) Cathodes for Microbial Fuel Cell Application. Bioresour. Technol. 2014, 163, 54–63, doi:10.1016/j.biortech.2014.03.091.
R6. Aksoylu, A.E.; Madalena, M.; Freitas, A.; Pereira, M.F.R.; Figueiredo, J.L. The Effects of Different Activated Carbon Supports and Support Modifications on the Properties of Pt/AC Catalysts. Carbon 2001, 39, 175–185, doi:10.1016/S0008-6223(00)00102-0.
Comment: b. Several plots with the results are termed: “Kinetics plots” or “Kinetics Data”. It is true that these are primary data that could be used to perform kinetic rate calculations; however, as presented they are simply concentration profiles of the contaminant degradation (measured in TOC, for example) vs time under certain conditions. I would suggest avoiding using the term related to kinetic data or kinetic plots to avoid confusion.
Response: Thank you for your valuable suggestion. We have changed the mentioned terms to the “plots”, “concentrations”, and “concentration profiles”. Please find the revised version of the manuscript.
Comment: 3. Caption of Figure 5: The co-authors should add more data to the caption: Purging of what? for example and indicate temperature and other relevant constant conditions. A similar situation is found for the Caption of Figure 3 where key conditions of the experiment/s are not indicated.
Response: Thank you for your valuable suggestion. We certainly agree that specification of reaction conditions is important. All the reaction parameters were mentioned in the Materials and Methods section and other parts of manuscript. The initial temperature of the reaction solutions was found to be 22 ± 1 °C, as shown in Figure S1 (Supporting Information).
According to your suggestion, we have added the value of initial temperature to the Materials and Methods section and modified the captures of mentioned figures. The revised version of manuscripts reads as follows:
LL. 187–188: Figure 3. Effect of the EG:H2O2 molar ratio on the TOC removal rate during EG degradation (initial EG concentration is 500 mg L–1, initial temperature is 22 ± 1 °C).
LL. 229–230: Figure 5. Comparison of EG mineralization rates under different purging conditions of the reaction medium (initial EG concentration is 500 mg L–1, initial temperature is 22 ± 1 °C).
LL. 363–364: The initial temperature of the reaction medium was 22 ± 1 °C (see Figure S1).
Comment: 4. Figure 8: It should be useful to add a picture of the laboratory apparatus set up to complement the illustration.
Response: Thank you for your valuable suggestion. We certainly agree that the addition of a picture of the laboratory setup in some cases can facilitate readers' understanding of the experimental technique. Our laboratory reactor is thermally insulated and is housed in the non-transparent casing. Sampling was done manually with a syringe, no additional instruments were used for the reaction except for a magnetic stirrer. It is worth noting that Figure 8(a) contains a comprehensive scheme of the reactor. The inset in Figure 8(b) shows a photo of the reactor equipped with mercury lamp in a quartz tube positioned using a lab-made holder.
According to your suggestion, we have added a photo of the experimental set-up and captions of the corresponding elements to the Supporting Materials. The revised manuscript reads as follows:
LL. 363–364: The exterior of the experimental setup is shown in Figure S7.
Supporting Information:
7. Experimental setup
As stated in the Materials and Methods section, the kinetic experiments on EG degradation upon addition to H2O2 under activation with UV-C light were carried out in a 1.5 L glass vessel with an inner-placed quartz tube, equipped with a 10 W germicidal lamp. In some experiments, the reaction medium was purged with air or oxygen through the PTFE tube. The reaction medium was stirred magnetically during the experiments. Figure S7 shows a photograph of the experimental setup used in the study.
Figure S7. Photo of the experimental setup used for the EG degradation.
Comment: 5. Figure 9 should be replaced by a small table possibly horizontal to avoid a large “blank” space in the reporting data.
Response: We used this plot specifically for better readers’ understanding the efficiency of the proposed solution to the problem of COD determination at the presence of H2O2. This plot clearly shows that in the whole range of H2O2 concentration the deviation of experimental value from the theoretical value is negligible in comparison to the entire scale. Therefore, we insist on showing these results in a form of plot in Figure 9.
According to your suggestion, we have added a mean value of the COD and its standard deviation and modified Figure 9 for better reading. The revised manuscript reads as follows:
LL. 402–408: Notably, the COD values measured for the samples pretreated with MnO2 to remove H2O2 residue (630 ± 6 mg L–1) corresponded well with the theoretical values regardless of the amount of H2O2 added over the whole studied range (Figure 9b).
Figure 9. (a) Absorbance spectra of the quinoneimine dye formed in the Trinder method (the inset shows the calibration plot for the determination of H2O2 concentration); (b) COD values measured for MnO2-pretreated EG solutions (500 mg L–1) containing different amounts of added H2O2.
Comment: 6. Some improvement in the English language would be beneficial although the majority of the draft is well written, and it has a nice reading flow. Examples of cases include the addition of the article “the” in front of the words efficiency and concentration (line 273); process (line 280); oxidation (line 287). These are some examples and there may be other similar cases.
Response: Thank you for your careful review. We apologize for any negligence. We have revised all the concerns you mentioned. Additionally, we have checked our manuscript using Language Assessment Tool (LAT) provided by AJE® (https://www.aje.com/grammar-check/) to check the text and polish it. This tool shows 8.6/10 score (94th percentile of papers submitted to AJE) and indicates that the manuscript is well written and does not need extensive language editing:
Comment:
7. The co-authors should change the term kinetic data, kinetic plots in the text by the suggested one in item 2.b.
Response: Thank you for your valuable suggestion. We have changed the mentioned terms to other ones: “plots”, “concentrations”, and “concentration profiles”. Please find the revised version of the manuscript.

Reviewer 2 Report
Comments and Suggestions for Authors
The present manuscript demonstrates the kinetic parameters of ethylene glycol degradation in wastewater in presence of H2O2 under UV-C light. While the contributions of the study are valuable, there are areas that require improvement. Addressing the following comments in the revised version would render the paper suitable for publication.
1. Page 3, lines 87 and 96-97: The authors have mentioned water contamination with ethylene glycol treated with UV-activated H2O2 in various studies. However, they did not include their observations from those studies in the present manuscript. It would be beneficial if the authors could provide those details, highlight the research gaps, and explain how the present study effectively addresses those gaps.
2. page 4. Figure 1b suggests that partial oxidation of ethylene glycol occurred due to the impact of UV light. In this experiment, the authors should analyze the reaction mixture and report the intermediates formed from the partial oxidation of ethylene glycol under UV light without H2O2. This will provide a clearer understanding of the oxidation of ethylene glycol in the presence of UV light and H2O2.
3. page 5, line 180: “oxydation” should be “the oxidation”
4. Page 7. The reaction mechanism proposed for the oxidation of ethylene glycol (see Figure 6) is not clear. The mechanism suggests that ethylene glycol directly produces 2-hydroxyacetaldehyde in the first step, followed by the formation of glyoxal via C-C cleavage. The formed glyoxal further leads to the formation of formic acid. The authors should provide clear and detailed explanations of how this sequence is possible. Similarly, it is important to offer a comprehensive explanation of how intermediates are formed at various steps in the C2 pathway, ultimately leading to CO2 as the final product. Detailed explanations and by-products formed in each step should be useful for the clear understanding of the reaction mechanism.
Author Response
Response to the Reviewer #2:
Dear the Reviewer #2,
thank you very much for your positive feedback and valuable comments on our manuscript. Please find the detailed answers and made changes below.
Comment: Page 3, lines 87 and 96-97: The authors have mentioned water contamination with ethylene glycol treated with UV-activated H2O2 in various studies. However, they did not include their observations from those studies in the present manuscript. It would be beneficial if the authors could provide those details, highlight the research gaps, and explain how the present study effectively addresses those gaps.
Response: Thank you for your valuable suggestion. Not so many studies consider the degradation of ethylene glycol using H2O2/UV-C system. This system is commonly used as a model system to investigate the main intermediates of EG degradation [R1, R2] or to compare with S2O82-/UV‑C system [R3]. In summary, the published studies have shown the possibility on application of H2O2/UV-C system for complete mineralization of ethylene glycol and suggested possible routes of the process, although differing depending on each specific study. However, the optimal parameters on H2O2amount as the oxidizing agent, the kinetic plots of intermediates, and possible approaches to intensify the oxidation process to reduce operating costs were not studied, whereas these aspects are very important for broader practical application of this technique.
References:
R1. McGinnis, B.D.; Adams, V.D.; Middlebrooks, E.J. Degradation of Ethylene Glycol in Photo Fenton Systems. Water Res. 2000, 34, 2346–2354, doi:10.1016/S0043-1354(99)00387-5.
R2. Dietrick McGinnis, B.; Dean Adams, V.; Joe Middlebrooks, E. Degradation of Ethylene Glycol Using Fenton’s Reagent and UV. Chemosphere 2001, 45, 101–108, doi:10.1016/S0045-6535(00)00597-X.
R3. Priyadarshini, M.; Ahmad, A.; Das, I.; Ghangrekar, M.M.; Dutta, B.K. Efficacious Degradation of Ethylene Glycol by Ultraviolet Activated Persulphate: Reaction Kinetics, Transformation Mechanisms, Energy Demand, and Toxicity Assessment. Environ. Sci. Pollut. Res. 2023, 30, 85071–85086, doi:10.1007/s11356-023-27596-9.
According to your suggestion, we have deeper discussed the previous investigations and research gaps. The revised manuscript reads as follows:
- 97–107: The application of this method in practice requires determination of the optimal parameters of EG oxidation and search for the approaches to intensify EG oxidation to avoid excessive consumption of oxidizing agent. However, a few studies related to the process of EG oxidation in H2O2/UV‑C systems have been published. For example, McGinnis et al. [29,45] reported the oxidation of EG in H2O2/UV‑C, Fe2+/UV‑C, and H2O2/Fe2+/UV‑C systems. A pathway based on a sequence of oxidation steps was proposed in these studies for the EG degradation. The validity of the proposed EG oxidation pathway in the H2O2/UV‑C system was questioned in [46], where the authors suggested the existence of two parallel pathways of EG oxidation with the formation of oxalic acid and formic acid. Similar pathways were proposed in [35] for EG oxidation using UV‑activated persulfate.
Comment: page 4. Figure 1b suggests that partial oxidation of ethylene glycol occurred due to the impact of UV light. In this experiment, the authors should analyze the reaction mixture and report the intermediates formed from the partial oxidation of ethylene glycol under UV light without H2O2. This will provide a clearer understanding of the oxidation of ethylene glycol in the presence of UV light and H2O2.
Response: Thank you for your valuable suggestion. We have studied the intermediates formed during UV-C treatment of EG solution and added discussion on these results to the text. The revised manuscript reads as follows:
- 141–145: Glycol aldehyde, acetaldehyde, formaldehyde, and formic acid were identified as the main oxidation products formed under these conditions. Figure S2 in the Supporting Information shows the concentration profiles of EG and formed products to illustrate changes in the composition of the reaction medium during UV‑C irradiation of the EG solution.
Supporting Information:
- Products of the EG oxidation during UV‑C treatment
The reduction of COD during the UV‑C irradiation with the rate of 36 mg L-1 h-1 was noticed. This fact allows us to conclude that partial oxidation of EG in this solution occurs. To determine the main oxidation products, the reaction solution was analyzed via HPLC. According to the results shown in Figure S2, the EG concentration reduces from 8.05 mM to 6.51 mM after 3 h of irradiation (Figure S2a) The main products of EG photolysis are glycolaldehyde, acetaldehyde, formaldehyde, and formic acid (Figure S2b). The results obtained correlate well with the literature data [1–3]. Aldehydes can be readily formed during the VUV irradiation of the ethylene glycol solution even in deoxygenated medium because of dehydrogenation. Formation of acetaldehyde can be explained by the dehydration of the HO-•CH-CH2-OH radical [3]. The C-C bond cleavage results in forming formaldehyde and formic acid. It can be concluded from the differences in EG kinetics that the formation of organic radicals without H2O2 occurs with a low rate. The presence of H2O2 and consequently the formation of OH• radicals lead to a greater rate of EG and byproducts oxidation.
Figure S2. Concentration profiles of (a) EG and (b) oxidation products formed during the treatment of the EG solution with UV-C.
Comment: page 5, line 180: “oxydation” should be “the oxidation”
Response: Thank you for your careful review. We apologize for any negligence. We have revised all the concerns you mentioned. Additionally, we have checked our manuscript using Language Assessment Tool (LAT) provided by AJE® (https://www.aje.com/grammar-check/) to check the text and polish it. This tool shows 8.6/10 score (94th percentile of papers submitted to AJE) and indicates that the manuscript is well written and does not need extensive language editing:
Comment: Page 7. The reaction mechanism proposed for the oxidation of ethylene glycol (see Figure 6) is not clear. The mechanism suggests that ethylene glycol directly produces 2-hydroxyacetaldehyde in the first step, followed by the formation of glyoxal via C-C cleavage. The formed glyoxal further leads to the formation of formic acid. The authors should provide clear and detailed explanations of how this sequence is possible. Similarly, it is important to offer a comprehensive explanation of how intermediates are formed at various steps in the C2 pathway, ultimately leading to CO2 as the final product. Detailed explanations and by-products formed in each step should be useful for the clear understanding of the reaction mechanism.
Response: Thank you for your valuable suggestion. Throughout the manuscript, we use the term “pathways” in relation to the EG oxidation process instead of “mechanism” to avoid any misunderstanding, because in this study we were not able to identify all the species (including radicals) formed in each step of EG oxidation. The scheme of EG oxidation and sequence of oxidation products are proposed based on the kinetic data (i.e., concentration profiles) on major intermediates, which were detected using HPLC and GC-MS. However, the formation of different esterification and etherification products can also be expected during this process [R1]. The proposed scheme of EG degradation agrees with the previous published papers [R1, R2], which describe two pathways of EG oxidation though the formation of formic and oxalic acids, respectively. The formation of CO2 as the final product can be expected because we detect a monotonic decrease in TOC value during the irradiation that means the complete oxidation (i.e., mineralization) of EG. As shown previously [R1, R3, R4], CO2 is the main product during EG mineralization.
References:
R1. Priyadarshini, M.; Ahmad, A.; Das, I.; Ghangrekar, M.M.; Dutta, B.K. Efficacious Degradation of Ethylene Glycol by Ultraviolet Activated Persulphate: Reaction Kinetics, Transformation Mechanisms, Energy Demand, and Toxicity Assessment. Environ. Sci. Pollut. Res. 2023, 30, 85071–85086, doi:10.1007/s11356-023-27596-9.
R2. Santos, L.; Lima Poli, A.; Schmitt Cavalheiro, C.; Neumann, M. The UV/H2O2 - Photodegradation of Poly(Ethyleneglycol) and Model Compounds. J. Braz. Chem. Soc. - JBCS 2009, 20, doi:10.1590/S0103-50532009000800012.
R3. Dietrick McGinnis, B.; Dean Adams, V.; Joe Middlebrooks, E. Degradation of Ethylene Glycol Using Fenton’s Reagent and UV. Chemosphere 2001, 45, 101–108, doi:10.1016/S0045-6535(00)00597-X.
R4. Rafieyan, S.G.; Marahel, F.; Ghaedi, M.; Maleki, A. Degradation of Mono Ethylene Glycol Wastewater by Different Treatment Technologies for Reduction of COD Gas Refinery Effluent. Int. J. Environ. Anal. Chem. 2024, 104, 3956–3975, doi:10.1080/03067319.2022.2098474.
According to your suggestion, we have deeper discussed the pathways and sequence of main oxidation products during EG degradation. The revised manuscript reads as follows:
- 267–282: The presence of glyoxal and the increase in formic acid concentration in the reaction medium were not observed after 2 h of irradiation, but changes in the concentrations of glycolaldehyde, glycolic acid, glyoxylic acid, and oxalic acid continued. From these results, it can be concluded that the oxidation of glycolic acid likely proceeded, with the formation of glyoxylic and oxalic acids only. The results obtained confirm previously observed effects, such as the absence of formic acid formation long after the beginning of irradiation during the EG oxidation process in the H2O2/UV‑C system [46]. Similar to the results above, the formation of oxalic acid was observed after a long irradiation time, in contrast to formic acid; therefore, the oxidation of EG includes two pathways involving the formation of formic and oxalic acids.
In the process of EG oxidation, the first step can be expected as an attack of EG by OH• radical with the formation of HO–•CH–CH2–OH and, subsequently, glycol aldehyde (Figure S3). However, it is difficult to predict a detailed mechanism of further radical transformations due to the presence of various reactive oxygen species [55,56]. The main pathways of EG oxidation process can be proposed on the basis of the concentration profiles of the oxidation intermediates.
- 287–291: This sequence is proposed on the basis of the observed sharp increase in the glycolaldehyde and formic acid concentrations, coupled with a slight increase in the glyoxal concentration during the first 0.5 h of irradiation. No significant differences in the concentrations of glyoxylic acid or oxalic acid were observed during the first 0.5 h of irradiation.
Supporting Information:
- Mechanism of EG oxidation by H2O2
As found in this study, the presence of H2O2 during the treatment of EG aqueous solution using UV‑C irradiation results in greater oxidation rate. Such an effect is a consequence of the formation of OH• radicals during H2O2 photolysis. These reactive oxygen species may be scavenged by EG triggering a formation of organic radicals and their further recombination and transformation (Figure S3). Thus, glycol aldehyde can be expected as a very first product of the EG oxidation in the H2O2/UV‑C system. Its further transformations may result in glyoxal and glycolic acid depending on the carbon atom that is attacked by •OH radicals. However, it is difficult to predict a detailed mechanism of such radical transformations [4,5].
Figure S3. First steps of EG oxidation with OH• radicals as main reactive species.

Round 2
Reviewer 1 Report
Comments and Suggestions for Authors
Please see attached document.

Author Response
Dear the Reviewer #1,
thank you very much for your positive feedback on our work and your suggestions for improving the quality of our manuscript.
Comment: A- In general, the co-authors introduced editing that seems to help with the reading flow of the manuscript and adjust the text as required.
Response: Thank you very much for your positive feedback and valuable comments, which were highly insightful and enable us to substantially improve the quality of this manuscript.
Comment: B- However, the most problematic aspect is the format: Unfortunately, the co-authors did not follow the recommended list of sections suggested in my previous review. This aspect, in the opinion of this reviewer, still needs to be adjusted. The reading of the MS is not in the correct sequence and, therefore, the audience may find it difficult to follow.
Response: We really appreciate your intention to improve the reading experience of the submitted manuscript, but as was confirmed by the Section Managing Editor, the section order of our manuscript is in accord with the Journal style and completely corresponds to all the Instructions for Authors provided by the Molecules journal (https://www.mdpi.com/journal/molecules/instructions). We believe our manuscript brings to light new data on the degradation of ethylene glycol as an emerging contaminant; thus, it can be useful to a broad readership for the design of efficient H2O2/UV-C oxidation systems for wastewater treatment.
Reviewer 2 Report
Comments and Suggestions for Authors
The authors of this manuscript have addressed all of my questions and incorporated their responses into the revised manuscript. I believe the revisions have significantly improved the manuscript, making it suitable for publication.
Author Response
Dear the Reviewer #2,
thank you very much for your positive feedback and valuable comments, which were highly insightful and enable us to substantially improve the quality of this manuscript.